# Thermal Stability and Degradation Kinetics of Patulin in Highly Acidic Conditions: Impact of Cysteine

**DOI:** 10.3390/toxins13090662

**Published:** 2021-09-16

**Authors:** Enjie Diao, Kun Ma, Hui Zhang, Peng Xie, Shiquan Qian, Huwei Song, Ruifeng Mao, Liming Zhang

**Affiliations:** 1Jiangsu Collaborative Innovation Center of Regional Modern Agriculture & Environmental Protection, College of Life Science, Huaiyin Normal University, Huai’an 223300, China; makun5060@163.com (K.M.); 8201611034@hytc.edu.cn (P.X.); 8201811112@hytc.edu.cn (S.Q.); 8200111016@hytc.edu.cn (H.S.); 8201711047@hytc.edu.cn (R.M.); 2Jiangsu Key Laboratory for Food Safety & Nutrition Function Evaluation, College of Life Science, Huaiyin Normal University, Huai’an 223300, China; 3Department of Food Science, College of Food Science & Engineering, Shandong Agricultural University, Tai’an 271018, China; 4Research & Development Center of National Vegetable Processing Technology, Jiangsu Liming Food Group Co., Ltd., Pizhou 221354, China; lmfood@163.com

**Keywords:** patulin (PAT), cysteine (CYS), degradation kinetics, thermal stability, Arrhenius equation

## Abstract

The thermal stability and degradation kinetics of patulin (PAT, 10 μmol/L) in pH 3.5 of phosphoric-citric acid buffer solutions in the absence and presence of cysteine (CYS, 30 μmol/L) were investigated at temperatures ranging from 90 to 150 °C. The zero-, first-, and second-order models and the Weibull model were used to fit the degradation process of patulin. Both the first-order kinetic model and Weibull model better described the degradation of patulin in the presence of cysteine while it was complexed to simulate them in the absence of cysteine with various models at different temperatures based on the correlation coefficients (*R*^2^ > 0.90). At the same reaction time, cysteine and temperature significantly affected the degradation efficiency of patulin in highly acidic conditions (*p* < 0.01). The rate constants (*k_T_*) for patulin degradation with cysteine (0.0036–0.3200 μg/L·min) were far more than those of treatments without cysteine (0.0012–0.1614 μg/L·min), and the activation energy (*E_a_* = 43.89 kJ/mol) was far less than that of treatment without cysteine (61.74 kJ/mol). Increasing temperature could obviously improve the degradation efficiency of patulin, regardless of the presence of cysteine. Thus, both cysteine and high temperature decreased the stability of patulin in highly acidic conditions and improved its degradation efficiency, which could be applied to guide the detoxification of patulin by cysteine in the juice processing industry.

## 1. Introduction

Patulin (PAT) is a low-molecular-weight mycotoxin that presents many hazards to the health of humans and animals based on the animal and cytotoxicity experiments. These hazards include agitation, vomiting, intestinal hemorrhages, convulsions, ulceration, nausea, and lesions in the duodenum, as well as chronic neurotoxic effects, such as genotoxic, teratogenic, and immunotoxic effects [1,2,3,4]. Patulin is produced by a number of different molds, such as *Penicillium*, *Aspergillus and Byssochlamys* [1,2,5]. *Penicillium expansum* is the main strain to produce patulin [6,7]. It can be found in damaged or moldy fruits, particularly apples. If contaminated apples are used to make juices, high levels of patulin are likely to be carried through to the final product [8,9,10]. Therefore, it is a potential health risk to consumers who often eat patulin-contaminated foods [1,2]. The World Health Organization (WHO) and the European Union (EU) established the maximum permitted levels of patulin in apple-based foods to protect consumer health, such as 50 μg/L in apple juice and cider, 25 ng/g in solid apple products, and 10 ng/g in products for infants and young children [11].

To remove or degrade patulin from contaminated foods, physical, chemical, and biological methods with various advantages and disadvantages have been developed by researchers to prevent patulin production or to remove or degrade it in agricultural products and foods for providing food safety [3,12]. Presently, no ideal method has been found to remove or degrade patulin in contaminated foods based on our investigation [12].

Cysteine (CYS) is an amino acid containing a thiol group (-SH) with nucleophilic properties, while patulin is an unsaturated γ-lactone with electrophilic properties, which can react to form PAT-CYS adducts based on the Michael addition reaction, a nucleophilic addition reaction at weak-acidic or near-neutral environments (pH 6.0–7.4) (Figure 1) [13,14]. Their toxicities are significantly lower than that of patulin itself [15,16]. In addition, cysteine is often used in food processing and storage as an anti-aging, anti-browning, radical scavenger, and reducing agent [17,18,19,20,21]. Therefore, it is maybe an ideal method in the detoxification of patulin in foods with weak-acidic or near-neutral pH values using cysteine. However, since the pH value of apple juice ranges from 3.0 to 4.0, it is difficult to form the PAT-CYS adducts based on the Michael addition reaction between them under these highly acidic conditions (pH < 6.0). Preliminary experiments found that high-temperature heating (>80 °C) can promote patulin degradation by cysteine under high-acid conditions. The aim of this study was to investigate the thermal stability and degradation kinetics of patulin in the absence and presence of cysteine in highly acidic conditions and to predict its degradation with or without cysteine.

Many degradation processes based on the chemical reactions in foods during thermal processing follow fixed-order kinetics, and the degradation kinetic order is defined by the power *n* in the isothermal degradation rate equation
(1)$$\frac{dC_{t}}{dt}=-k_{T}C_{t}^{n}$$
where $C_{t}$ is the concentration of patulin at time t, and $k_{T}$ is a temperature-dependent rate constant. In principle, once the degradation kinetic order *n* and $k_{T}$ have been determined, one can estimate and predict the concentration of patulin in a reaction system when the temperature is a function of time. Under the isothermal conditions, Equation (1) can be integrated to obtain the following order kinetics due to the rate constant $k_{T}$: when the kinetic order *n* = 0,
(2)$$C_{t}=C_{0}-k_{T}t$$
it is a zero-order kinetics; when *n* = 1,
(3)$$C_{t}=C_{0}\;exp\left(-k_{T}t\right)$$
it is the first-order kinetics; when *n* = 2
(4)$$\frac{1}{C_{t}}=\frac{1}{C_{0}}+k_{T}t$$
it is the second-order kinetics; and for the *n*th kinetics (*n* > 2),
(5)$$C_{t}={\left(C_{0}^{1-n}+k_{T}\left(n-1\right)t\right)}^{1/\left(1-n\right)}$$
where $C_{0}$ is the initial concentration of patulin at time zero.

In addition, the Weibull model was used to simulate the degradation process of patulin:(6)$$\frac{C_{t}}{C_{0}}=exp\left(-kt^{n}\right)$$
where *k* is the Weibull scale parameter (min^−*n*^), which is analogous to a rate constant dependent on reaction temperature, and *n* is the shape parameter describing the shape of the degradation curve. When *n* = 1, Equation (6) is equivalent to a first-order kinetics (Equation (3)), and when *n* ≠ 1, the degradation rate increases (*n* > 1) or decreases (*n* < 1) with time. The shape parameter *n* remains constant with increasing temperature if the reaction mechanism is not affected by temperature.

The Arrhenius equation was used to compare the temperature dependency of patulin degradation by cysteine:(7)$$k_{T}=A\;exp\left(-\frac{E_{a}}{R}\times \frac{1}{T}\right)\;\mathrm{or}\;ln\left(k_{T}\right)=-\frac{E_{a}}{R}\times \frac{1}{T}+lnA$$
where $E_{a}$ is the activation energy (J/mol), *R* is the gas constant (8.314 J/mol·*K*), *T* is the absolute temperature (*K*), and *A* is the exponential factor that is assumed to be independent of temperature.

## 2. Results

### 2.1. Thermal Stability of Patulin in Highly Acidic Conditions

To investigate the thermal stability of patulin in a highly acidic environment, the acidic condition of apple juice was simulated at pH 3.5 in a PCA buffer solution. The result is shown in Figure 2.

As seen in Figure 2, heat treatment at 90 °C for 30 min decreased patulin from 125.16 ± 0.53 μg/L to 121.25 ± 8.89 μg/L and only decreased by 3.13% (*p* > 0.05). Compared to the control, the degradation efficiency of patulin significantly increased to 18% at 90 °C for 60 min (*p* < 0.05), while no obvious changes happened to the degradation efficiencies of patulin at 90 °C for 90–180 min (*p* > 0.05). Similarly, the degradation efficiencies of patulin were only 4.34% and 4.19% at 120 and 150 °C for 30 min (*p* > 0.05), respectively. Overall, the degradation efficiency of patulin improved with the increase in temperature when the reaction time was greater than 30 min (*p* < 0.01). After 180 min of heat treatment, the degradation efficiencies of patulin reached 22.75% and 47.14% at 90 and 120 °C, respectively, and it completely disappeared at 150 °C.

### 2.2. Effect of Cysteine on the Degradation Efficiency of Patulin

In this study, adding cysteine did significantly decrease the thermal stability of patulin, and then its degradation efficiency improved at temperatures greater than 90 °C in PCA buffer solution with pH 3.5 (Figure 2A). The degradation efficiency of patulin by cysteine notably increased with the increases in temperature and reaction time (*p* < 0.01, Figure 2B). Compared to the degradation efficiencies of patulin without cysteine, they reached 24.50, 37.97, and 54.49% at 90, 120, and 150 °C for 30 min, respectively (*p* < 0.01). After heat treatment for 90 min at the same temperature (90–150 °C), the corresponding degradation efficiencies of patulin were 42.12, 78.94, and 100%, which were about twice those of treatments without cysteine.

### 2.3. Degradation Kinetic Models of Patulin with and without Cysteine

According to the results above mentioned, the temperature, reaction time, and presence of cysteine all significantly affected the degradation efficiency of patulin in high-acid conditions. The zero-, first-, and second-order kinetic models were used to fit to all experimental data with and without cysteine. Figure 3 shows the degradation of patulin without cysteine and the fitted kinetic models at different temperatures (90–150 °C). Table 1 lists the corresponding kinetic equations, rate constants (*k_T_*), and correlation coefficients (*R*^2^). *R*^2^ is an indicator of the strength of the linear relationship between two different variables, and a correlation close to 1.0 indicates a perfect positive correlation. Taking into account to the highest value of *R*^2^, the second-order kinetic model best described the patulin degradation by heat treatment at 90 °C, and the first-order kinetic model best fit it at 150 °C without cysteine. The zero-, first-, and second-order kinetic models all best simulated the process of patulin degradation at 120 °C without cysteine due to the very close *R*^2^ values (0.9400–0.9449) (Table 1). In addition, it can be seen from Table 1 that for each kinetic model, the higher the temperature, the greater the value of *k_T_*.

Likewise, in the presence of cysteine, both the zero- and first-order kinetic models best described the degradation of patulin at 90 °C based on the high *R*^2^ (0.9634–0.9652), and the first-order kinetic model best simulated them at 120 °C (*R*^2^ = 0.9910) and 150 °C (*R*^2^ = 0.9300), respectively (Table 2 and Figure 4). Similarly, the higher the temperature, the greater the value of *k_T_* for each kinetic model (Table 2).

Figure 5 illustrates the fitted curves of the Weibull models to the data for degradation of patulin with and without cysteine in pH 3.5 of PCA buffer solutions. The degradation efficiency of patulin in the presence of cysteine at different temperatures complied well with the Weibull model due to the non-linear decrease in *C_t_*/*C*_0_ with the reaction time (*R*^2^ = 0.9297–0.9910) (Figure 5B and Table 3). The degradation efficiency of patulin without cysteine also followed the Weibull model at 120 and 150 °C (*R*^2^ = 0.9227–0.9450) (Figure 5A and Table 3). The *k_T_* increased with the increase in temperature ranging from 90 to 150 °C.

Figure 6 depicts the degradation of patulin without and with cysteine based on the Arrhenius equation at different temperatures. Seen from the Figure 6, the relationship between *ln*(*k_T_*) and the inverse of the absolute temperature (1/*T*) were highly linear (*R*^2^ = 0.9755 without cysteine and 0.9986 with cysteine), respectively, which showed an Arrhenius-type temperature dependency for both reactions. The shape parameter *n* = 1, which remained with increasing temperature, indicating the reaction mechanism between patulin and cysteine, was not affected by temperature. According to Equation (7), the calculated activation energies (*E_a_*) for the degradation of patulin without and with cysteine were 61.74 kJ/mol and 43.89 kJ/mol, respectively.

## 3. Discussion

### 3.1. Thermal Stability of Patulin without and with Cysteine

Patulin is very stable to heat, especially in highly acidic conditions. The half-life of patulin in Sorensen’s phosphate buffer is 64 h at pH 8 while 1310 h at pH 6 [22,23]. It is reported that patulin is quite stable at temperatures ranging from 105 to 125 °C in aqueous solutions with pH 3.5–5.5, and it gradually becomes unstable as the pH increases [23]. Heat treatments at 90 and 100 °C for 20 min only decreased by 18.81 and 25.99% of the initial patulin concentration in apple juice, respectively, and the corresponding values were only 9.40 and 14.06% for 70 and 80 °C evaporation, respectively [24]. At pH 6.0, patulin did not significantly decrease at 50 °C for 20–120 min and decreased by about 50% at 100 °C for 40–60 min. It was also reported that patulin was very stable in acidic media (pH < 5.0) and quickly decomposed at pH 6 [25]. Similarly, the mean loss of patulin in apple juice was 39.6% after pasteurization at 90 °C for 30 s [26]. These results demonstrated the thermal stability of patulin to different temperatures in an acidic environment (pH < 6.0). Thus, it is very difficult to completely remove or degrade patulin during the processing stages of apple juice. In this study, the thermal stability of patulin was furtherly verified in pH 3.5 of the PCA buffer solution without cysteine. When the temperature was less than 90 °C, it was difficult to degrade patulin in highly acidic conditions by heat processing. In other same treatment conditions, adding cysteine could obviously decrease the thermal stability of patulin, and this trend gradually increased with the increase of temperature.

It is well known that patulin is an unsaturated γ-lactone, which can react with L-cysteine based on the Michael addition reaction, a nucleophilic addition reaction, and form PAT-CYS adducts [13,14]. The toxicities of these adducts are significantly lower than that of patulin itself [15,16]. However, the Michael addition reaction (a nucleophilic addition reaction) is carried out spontaneously under weak-acidic or near-neutral conditions (pH 6.0–7.4) [14,16]. The pH value of apple juice ranges from 3.0 to 4.0, so it is difficult to form the PAT-CYS adducts based on the Michael addition reaction between patulin and cysteine under highly acidic conditions (pH < 6.0). In this study, cysteine still effectively degraded patulin in highly acidic conditions (pH 3.5) at temperatures greater than 90 °C, while the degradation mechanisms of patulin by cysteine were still unknown. However, we can infer that there are three possible mechanisms leading to the effective degradation of patulin by cysteine under high acid and high temperature based on the reported literatures [14,25]. The first mechanism remains a nucleophilic addition reaction. Increasing the temperature significantly increases the motion speed of patulin and cysteine molecules in the solution, and then increases the chance for mutual collisions between the active sites (-SH, -NH_2_, and -COOH groups) of cysteine with patulin, thus facilitating the nucleophilic addition reaction between them [14]. The second one is the combination of thermal degradation and nucleophilic addition reaction. According to the previous research [25], patulin is firstly thermo-degraded to 3-keto-5-hydroxypentanal and glyoxylic acid at high temperatures (>80 °C), which might be a reversible process. Both products contain an aldehyde group with high activation and electrophilic properties, which can react with cysteine (nucleophilic agent) to form their adducts (Figure 7). The third is forming hydrogen bonds and covalent bonds between patulin and cysteine (-SH and -NH_2_ groups) in high-acid and high-temperature environments [27,28,29]. These inferred mechanisms should be further verified based on the analysis of Fourier transform infrared spectra (FTIR), mass spectroscopy (MS), nuclear magnetic resonance (NMR), and thin-layer chromatography (TLC) in the next work.

### 3.2. Thermal Degradation Kinetic Models of Patulin

The degradation kinetic models of patulin were various with the different temperatures in high-acid conditions, as well as the absence and presence of cysteine. According to the results from this study, various models could fit the degradation processes of patulin well under the same reaction conditions based on the high *R*^2^ values. Compared to the *k_T_* values at the same temperature, it was easily found that the *k_T_* values of treatments with cysteine were all notably greater than those without cysteine, indicating the importance of cysteine in degrading patulin in highly acidic conditions. For the same kinetic model, increasing temperature also significantly increased the *k_T_* value, regardless of the presence of cysteine, showing the critical role of temperature in the degradation of patulin. Similar conclusions were obtained in the previous study [30]. In addition, the lower *E_a_* for the degradation of patulin in the presence of cysteine is consistent with the promoting role of cysteine reported in this study, and perhaps the two reactions were performed through different mechanisms.

## 4. Conclusions

Overall, patulin is very stable to heat in pH 3.5 of the PCA buffer solution when temperatures are less than 90 °C. The degradation efficiency of patulin without cysteine significantly improved with the increase in reaction time when the temperature exceeded 120 °C. In the presence of cysteine, the degradation efficiency of patulin at the same temperature notably improved, indicating its important role in patulin degradation. The rate constant (*k_T_*) of patulin degradation with cysteine was far more than that of treatments without cysteine at the same temperature, and the corresponding activation energy (*E_a_*) was also far less than that of treatments without cysteine. Thus, the temperature, reaction time, and presence of cysteine all significantly affected the thermal stability of patulin and promoted its degradation in high-acid conditions. Both the first-order model and Weibull model best described the degradation of patulin because the correlation coefficients (*R*^2^) were all greater than 0.90, while it was complex to elaborate the degradation process of patulin in the absence of cysteine. At the temperature of 90 °C, no models could better fit the degradation process of patulin due to the *R*^2^ less than 0.85, while the zero-, first-, second-order and Weibull models could all better simulate it at 120 °C (*R*^2^ > 0.94). When the reaction temperature was at 150 °C, both the first-order model and Weibull model could better describe its degradation with *R*^2^ > 0.92. Therefore, cysteine can effectively degrade patulin in highly acidic conditions with the help of temperatures greater than 90 °C. However, the degradation mechanisms of patulin by cysteine in high-temperature and high-acid conditions are still unknown, their degradation products are not identified, and their safety has not been evaluated yet. Therefore, the resolution of these problems in the next work will help to promote the practical application of patulin degradation by cysteine in the juice processing industry.

## 5. Materials and Methods

### 5.1. Materials

Patulin (purity ≥ 98.0%) was purchased from Sangon Biotech (Shanghai, China) Co., and Ltd. L-cysteine (purity ≥ 99.0%) was purchased from the Aladdin Industrial Corporation (Shanghai, China). Acetonitrile (HPLC grade) was obtained from Oceanpak Alexative Chemical Co., Ltd. (Goteborg, Sweden). Formic acid (HPLC grade) was provided by Anpel Laboratory Technologies Inc. (Shanghai, China). Ethyl acetate, disodium hydrogen phosphate, and citric acid were all analytical grade and purchased from Sinopharm Chemical Reagent Co., Ltd. (Shanghai, China).

### 5.2. Methods

#### 5.2.1. Preparation of Patulin and Cysteine Solutions

To simulate the acidic condition of apple juice, phosphoric-citric acid (PCA) buffer solutions of pH 3.5 were firstly prepared by mixing 0.2 mol/L Na_2_HPO_4_ solution and 0.1 mol/L citric acid solution with suitable volumes, which were checked using a pH meter (pHs-3C, Leici, Shanghai, China) and adjusted with 1 mol/L of phosphoric acid or sodium hydroxide solution. A total of 154.12 μL of standard patulin solution (1.00 mg/mL acetonitrile solution) was diluted to 100 mL in a brown volumetric flask by PCA buffer solutions with pH 3.5 and made 10 μmol/L of working solution of patulin. Similarly, a working solution of cysteine (30 μmol/L) was obtained by diluting the standard solution of cysteine (1817.4 μL, 1 mg/mL) to 50 mL using a PCA buffer solution with pH 3.5.

#### 5.2.2. Reaction of Patulin and Cysteine

A total of 30 μmol/L of cysteine solutions and 10 μmol/L of patulin solutions with a molar ratio of 3:1 (pH 3.5) were added into a 10 mL reaction kettle and were quickly mixed and placed in an oil bath at different temperatures (90, 120, and 150 °C) to react for different times (0, 30, 60, 90, 120, 150, and 180 min), respectively. Then, the reaction solutions were cooled to room temperature with running water immediately. The samples were treated without cysteine in the same reaction conditions as the controls.

#### 5.2.3. Determination of Patulin in Reaction Solutions

Patulin was determined by the HPLC method. A total of 2 mL of the reaction solution, after being cooled to room temperature, and 5 mL of ethyl acetate were put into a 60 mL separating funnel and extracted by shaking vigorously for 5 min. The top layer of ethyl acetate was separated and evaporated to dryness with a rotary evaporator (Model RE-201C, KEHUA, Zhengzhou, China) at 40 °C with reduced pressure. The residue was dissolved with 10 mL of formic acid water (0.1%, *v/v*), which was filtrated using 0.22 μm of inorganic filter film. The filtrate was used for the determination of patulin by the HPLC system (Agilent 1260 with DAD detector, Palo Alto, CA, USA).

The analysis was performed under the following conditions: the detection wavelength was set at 276 nm; chromatographic column: Waters XBridge TM (100 nm × 4.6 mm, 3.5 μm C18 stationary phase); mobile phase was 0.1% formic acid water: acetonitrile = 95:5 (*v/v*) with a flow rate of 0.75 mL/min; the column temperature was at 30 °C; and the injection volume was 20 μL.

#### 5.2.4. Statistical Analysis

All thermal degradation assays were performed in triplicate, and the means ± standard deviation (SD) were used to simulate the kinetics. One-way ANOVA was carried out to determine any significant difference (*p* < 0.05) among the various treatment groups using SPSS 18.0 software (IBM, Chicago, IL, USA).

## Figures and Tables

**Figure 1 toxins-13-00662-f001:**
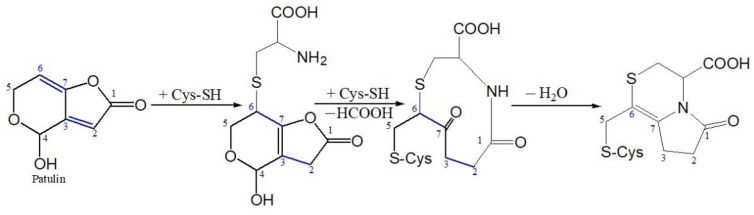
Michael addition reaction between patulin (PAT) and cysteine (CYS) suggested by Fliege and Metzler [14].

**Figure 2 toxins-13-00662-f002:**
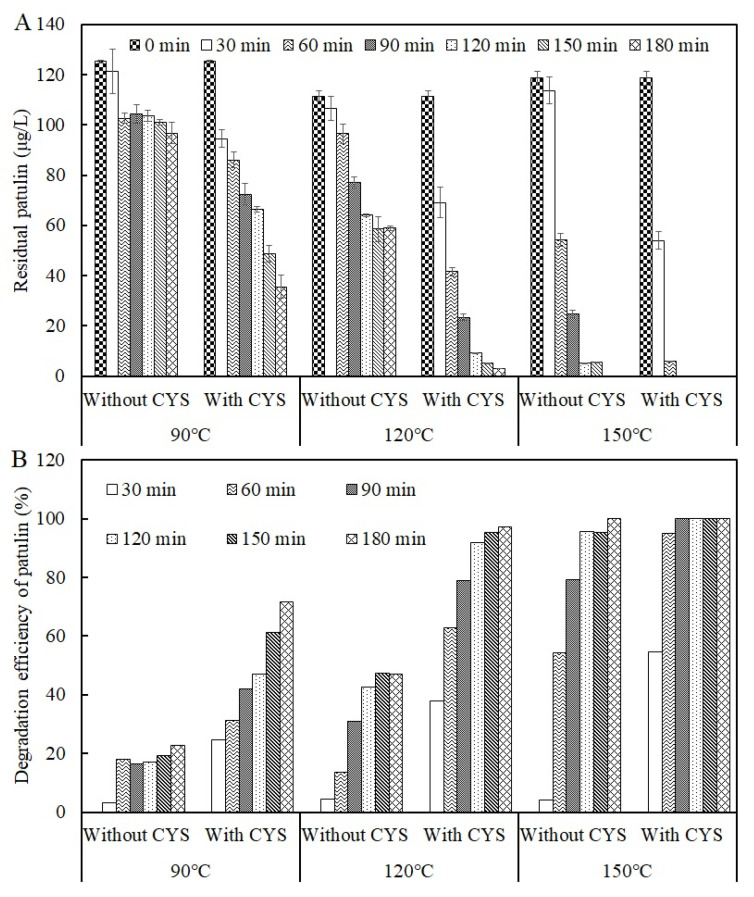
Residual concentrations (**A**) and degradation efficiencies (**B**) of patulin in pH 3.5 of solutions with and without cysteine at different temperatures (90–150 °C) (Appendix A).

**Figure 3 toxins-13-00662-f003:**
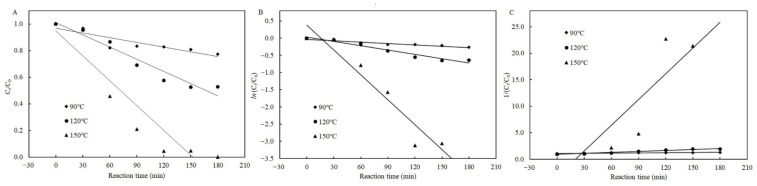
Plots of the zero- (**A**), first- (**B**), and second-order (**C**) kinetic models for patulin degradation in pH 3.5 of solutions without cysteine at different temperatures (90–150 °C) (Appendix A).

**Figure 4 toxins-13-00662-f004:**
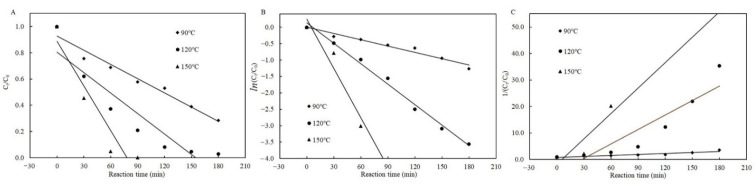
Plots of the zero- (**A**), first- (**B**), and second-order (**C**) kinetic models for patulin degradation in pH 3.5 of solutions with cysteine at different temperatures (90–150 °C) (Appendix A).

**Figure 5 toxins-13-00662-f005:**
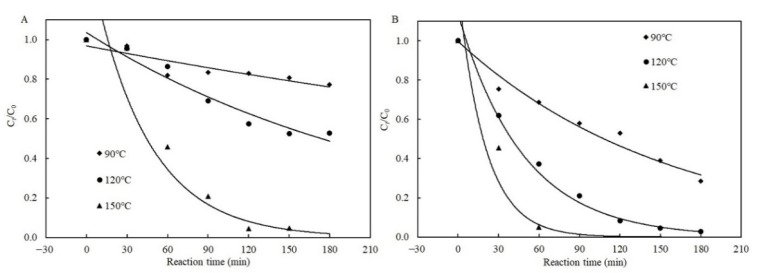
Plots of the Weibull kinetic models for patulin degradation in pH 3.5 of solutions without (**A**) and with cysteine (**B**) at different temperatures (90–150 °C) (Appendix A).

**Figure 6 toxins-13-00662-f006:**
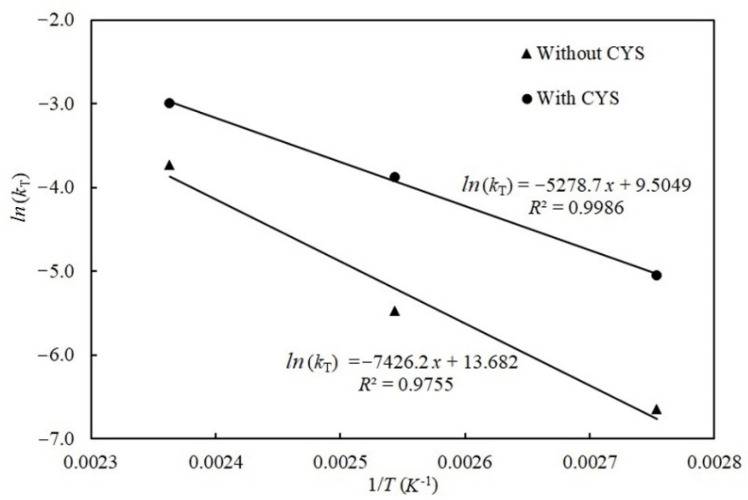
Arrhenius plot for patulin degradation in pH 3.5 of solutions without (▲) and with cysteine (●) at different temperatures (90–150 °C) (Appendix A).

**Figure 7 toxins-13-00662-f007:**
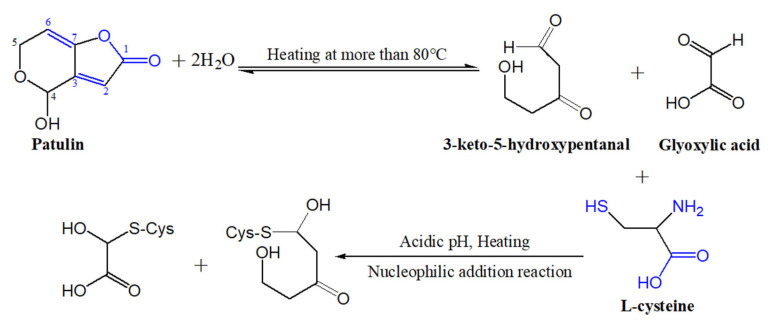
Suggested degradation mechanisms of patulin by cysteine at high-temperature and high-acid conditions.

**Table 1 toxins-13-00662-t001:** Comparison of the fitted zero-, first-, and second-order kinetic models to the experimental data for the degradation of patulin (PAT) without cysteine (CYS) at different temperatures (Appendix A).

Kinetic Model	Temperature (°C)	Kinetic Equation	*k_T_* (μg/L·min)	*R* ^2^
Zero-order	90	*C_t_*/*C*_0_ = −0.0012*t* + 0.9684	0.0012	0.7868
	120	*C_t_*/*C*_0_ = −0.0031*t* + 1.0095	0.0031	0.9400
	150	*C_t_*/*C*_0_ = −0.0062*t* + 0.9491	0.0062	0.8731
First-order	90	*ln*(*C_t_*/*C*_0_) = −0.0013*t* − 0.0328	0.0013	0.7988
	120	*ln*(*C_t_*/*C*_0_) = −0.0042*t* + 0.0349	0.0042	0.9449
	150	*ln*(*C_t_*/*C*_0_) = −0.0241*t* + 0.3802	0.0241	0.9228
Second-order	90	1/(*C_t_*/*C*_0_) = 0.0015*t* + 1.0339	0.0015	0.8108
	120	1/(*C_t_*/*C*_0_) = 0.0059*t* + 0.9220	0.0059	0.9424
	150	1/(*C_t_*/*C*_0_) = 0.1614*t* − 3.2556	0.1614	0.7711

**Table 2 toxins-13-00662-t002:** Comparison of the fitted zero-, first-, and second-order kinetic models to the experimental data for the degradation of patulin with cysteine at different temperatures (Appendix A).

Kinetic Model	Temperature (°C)	Kinetic Equation	*k_T_* (μg/L·min)	*R* ^2^
Zero-order	90	*C_t_*/*C*_0_ = −0.0036*t* + 0.9289	0.0036	0.9652
	120	*C_t_*/*C*_0_ = −0.0052*t* + 0.8039	0.0052	0.8662
	150	*C_t_*/*C*_0_ = −0.0114*t* + 0.8870	0.0114	0.9010
First-order	90	*ln*(*C_t_*/*C*_0_) = −0.0064*t* − 0.0023	0.0064	0.9634
	120	*ln*(*C_t_*/*C*_0_) = −0.0207*t* + 0.1271	0.0207	0.9910
	150	*ln*(*C_t_*/*C*_0_) = −0.0501*t* + 0.2379	0.0501	0.9300
Second-order	90	1/(*C_t_*/*C*_0_) = 0.0126*t* + 0.7976	0.0126	0.8805
	120	1/(*C_t_*/*C*_0_) = 0.1822*t* − 5.0421	0.1822	0.8307
	150	1/(*C_t_*/*C*_0_) = 0.3200*t* − 1.7988	0.3200	0.7968

**Table 3 toxins-13-00662-t003:** Comparison of the fitted Weibull kinetic models to the experimental data for the degradation of patulin without and with cysteine at different temperatures (Appendix A).

Kinetic Model	Temperature (°C)	Kinetic Equation	*k_T_* (μg/L·min)	*R* ^2^
Without cysteine	90	*C_t_*/*C*_0_ = 0.9683 *exp*(−0.001*t*)	0.001	0.7977
	120	*C_t_*/*C*_0_ = 1.0358 *exp*(−0.004*t*)	0.004	0.9450
	150	*C_t_*/*C*_0_ = 1.4633 *exp*(−0.024*t*)	0.024	0.9227
With cysteine	90	*C_t_*/*C*_0_ = 0.9971 *exp*(−0.006*t*)	0.006	0.9641
	120	*C_t_*/*C*_0_ = 1.1370 *exp*(−0.021*t*)	0.021	0.9910
	150	*C_t_*/*C*_0_ = 1.2694 *exp*(−0.050*t*)	0.050	0.9297

## Data Availability

The data that support the finding of this study are openly available online.

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
