# Peer review of "Thermal Stability and Degradation Kinetics of Patulin in Highly Acidic Conditions: Impact of Cysteine"

_toxins, 2021, doi:10.3390/toxins13090662_

Round 1
Reviewer 1 Report
The paper described the degradation kinetics of patulin with cysteine and heat. The subject area is of interest although there has been previous research in the area which should be included within the Introduction. The authors frequently refer to apple juice as a model but in reality only performed studies in phosphoric acid. Therefore, the only resemblance to apple juice was pH and in this respect it would have been interesting to assess the effect of ascorbic acid on the reaction. The study was superficial and should have at least confirmed the structure of adducts rather than speculating on mechanisms.
Specific points
The Abstract should be made more informative.
Line 5: What acid was used to adjust the pH? What was the range of cysteine concentrations applied?
Line 9: What was complexed?
Line 13: The authors should include degradation rates. Data should be compared in terms of statistical differences.
Line 27: Long sentence. What do the authors consider low molecular weight?
Line 43: What do the authors consider an ideal method? There have been methods described and these should be cited within the Introduction.
Line 50: Which food is cysteine used? In addition, the authors should comment of the stability of cysteine at alkali pH.
Line 55: Were the preliminary studies published? Is the Patulin-cystine adducts cytotoxic?
Line 88: The solution only represents apple juice with respect to pH. Therefore, phosphoric acid solutions would be more appropriate.
Figure 2: Actual concentration of cysteine would be more informative than % decline. The authors should consider highlighting statistical differences and error bars within the graph. The legend should have sufficient information so that the figure can be understood in isolation.
Line 105: What concentration of cysteine?
Line 171-183: This part belongs within the Introduction.
Line 187: Is it more that cystine adduct formation was enhanced with temperature rather than decreasing the thermal stability of the mycotoxin. It would have been interesting to compare the reaction of ascorbic acid and cystine.
Line 199: The authors could only speculate on the mechanisms and should have at least confirmed the structure of the adduct.
Line 267: Is phosphoric acid representative of apple juice? Malic acid would have been more appropriate or even apple juice if this was the system being represented.
Line 279: Would these temperatures represent those in apple juice processing?
Line 292: Was adduct formation determined by the disappearance of patulin or formation of Patulin-cysteine?
Author Response
Dear Reviewer:
Thank you for the comments concerning our manuscript entitled “Thermal Stability and Degradation Kinetics of Patulin in Highly Acidic Conditions: Impact of Cysteine” (ID: toxins-1364184). Those comments are all valuable and very helpful for revising and improving our paper. We have studied the comments carefully and made corrections which we hope to meet with approval. The main corrections in the paper and the responds to your comments are as follows:
(1) Comment: The paper described the degradation kinetics of patulin with cysteine and heat. The subject area is of interest although there has been previous research in the area which should be included within the Introduction. The authors frequently refer to apple juice as a model but in reality only performed studies in phosphoric acid. Therefore, the only resemblance to apple juice was pH and in this respect it would have been interesting to assess the effect of ascorbic acid on the reaction. The study was superficial and should have at least confirmed the structure of adducts rather than speculating on mechanisms.
Response: Thank you very much for the suggestions. The questions you asked were exactly what we had been considered before this study. According to the reported literatures (Harwig, Scott, Kennedy & Chen, 1973; Pohland & Allen, 1970; Scott & Somers, 1968), researchers found that patulin in some foods was unstable, and disappeared from these commodities due to its reaction with amino acids or protein containing thiol. They deduced that a Michael addition reaction involved this process, i.e. thiol group on amino acids or protein bonded to the double bond of the unsaturated lactone system of patulin (Ciegler, Beckwith & Jackson, 1976; Jones & Young, 1968). The Michael addition reaction can be defined as the nucleophilic addition of a nucleophile to an ?, ?-unsaturated carbonyl compound. Fliege & Metzler (2000) pointed out the nucleophilic addition reaction between patulin and nucleophiles based on the electrophilic properties of patulin (Fig.1). The nucleophiles, L-cysteine (CYS), reacted with patulin to form the covalent adducts, which were spontaneous happened at near-neutral pH, and the thiol group of the nucleophiles attached to one of the double bonds of patulin. However, it is very difficult for the nucleophilic addition reaction to happen under the strong acidic conditions due to the absence of the thiolate ions (Rodríguez-Bencomo et al., 2020). Fig.1 The nucleophilic addition reaction between patulin and nucleophiles based on the electrophilic properties of patulinOur pre-experiments also found that patulin is very stable to heat in strong acidic conditions, which is very difficult to remove or degrade by cysteine based on the Michael addition reaction (nucleophilic addition reaction) at room temperature. It is necessary to effectively remove patulin by cysteine with the help of high temperature greater than 90℃ and long reaction time more than 0.5 h (Fig.2 and Fig.3). Therefore, we think that the pH value of foods is one of the critical factors influencing the removal of patulin in foods. Apple juice as an apple based food, which is often contaminated by patulin. Its pH value is in the range from pH 3.0 to 4.5, and usually is around 3.5. In this study, we used pH 3.5 of phosphoric-citric acid (PCA) buffer solution to simulate the acidic condition of apple juice, and the main objectives are to investigate the thermal stability and degradation kinetics of patulin in highly acidic conditions with and without the cysteine. In order to clearly understand the effects of cysteine on the thermal stability of patulin in highly acidic conditions, and to reduce the interference of juice compositions, the study used PCA buffer solution to simulate the apple juice. In addition, ascorbic acid has been studied to remove or degrade patulin in mode system and apple juice by some researchers (Brackett & Marth, 1979; Kokkinidou, Floros, & Laborde, 2014; Ei Hajj Assaf er al., 2019). So this study has not assessed the effects of ascorbic acid on the reaction.Fig.2 Effect of solution pH on the removal of patulin by cysteine. (Reaction conditions: pH 3.0~7.0, molar ratio between cysteine and patulin 3:1, reaction temperature 120℃, and reaction time 120 min).Fig.3 Effect of reaction temperature on the removal of patulin by cysteine. (Reaction conditions: pH 3.5, molar ratio between cysteine and patulin 3:1, reaction time 120 min, and reaction temperature 30~150℃). About the structures of adducts, we think that they are not the focus of this study. In this study, the main objectives are to investigate the thermal stability and degradation kinetics of patulin in highly acidic conditions with and without the cysteine. The removal mechanisms of patulin by cysteine were only speculated by us based on the studies reported by Fliege & Metzler (2000) and Collin et al. (2008). In the discussion, it is only to explain the reasons about the removal of patulin by cysteine at highly acidic and high temperature conditions. References:
Brackett, R.E., Marth, E.H., 1979. Ascorbic acid and ascorbate cause disappearance of patulin from buffer solutions and apple juice. Journal of Food Protection 42(11): 864–866.
Ciegler, A., Beckwith, A.C. and Jackson, L.K., 1976. Teratogenicity of patulin and patulin adducts formed with cysteine. Applied and Environmental Microbiology 31(5): 664–667.
Collin, S., Bodart, E., Badot, C., Bouseta, A., & Nizet, S., 2008. Identification of the main degradation products of patulin generated through heat detoxification treatments. Journal of the Institute of Brewing 114(2): 167–171.
El Hajj Assaf C., De Clercq N., Van Poucke C., et al., 2019. Effects of ascorbic acid on patulin in aqueous solution and in cloudy apple juice. Mycotoxin Research 35(4): 341–351.
Fliege, R. and Metzler, M., 2000. Electrophilic properties of patulin. N-Acetylcysteine and glutathione adducts. Chemical Research in Toxicology 13: 373–381.
Harwig, J., Scott, P.M., Kennedy, B.P.C. and Chen, Y.K., 1973. Disappearance of patulin from apple juice fermented by Saccharomyces spp. Canadian Institute of Food Science and Technology Journal 6(1): 45–46.
Jones, J.B. and Young, J.M., 1968. Carcinogenicity of lactones. III. The reactions of unsaturated γ–lactones with L-cysteine. Journal of Medicinal Chemistry 11(6):1176–1182.
Kokkinidou, S., Floros, J.D., Laborde, L.F., 2014. Kinetics of the thermal degradation of patulin in the presence of ascorbic acid. Journal of Food Science 79(1), T108-T114.
Pohland, A.E. and Allen, R., 1970. Stability studies with patulin. Journal of Association of Official Analytical Chemists 53(4):688–691.
Rodríguez-Bencomo, J.J., Sanchis, V., Viñas, I., Martín-Belloso, O. and Soliva-Fortuny, R., 2020. Formation of patulin-glutathione conjugates induced by pulsed light: A tentative strategy for patulin degradation in apple juices. Food Chemistry 315: https://doi.org/10.1016/j.foodchem.2020.126283.
Scott, P.M. and Somers, E., 1968. Stability of patulin and penicillic acid in fruit juices and flour. Journal of Agricultural and Food Chemistry 16(3): 483–485.
(2) Comment: The Abstract should be made more informative.
Response: Thanks you for the good suggestion. We added some data (blue font) to make the abstract more informative and accessible to readers.Abstract: The thermal stability and degradation kinetics of patulin (10 μmol/L) in pH 3.5 of phosphoric-citric acid buffer solutions in the absence and presence of cysteine (30 μmol/L) were investigated at the temperatures ranged from 90 to 150℃. The zero-, first-, second-order models, and the Weibull model were used to fit the degradation process of patulin. Both the first-order kinetic model and Weibull model better described the degradation of patulin in the presence of cysteine, while it was complexed to simulate them in the absence of cysteine with various models at different temperatures based on the correlation coefficients (R2>0.90). At the same reaction time, cysteine and temperature significantly affected the degradation efficiency of patulin in highly acidic conditions (p<0.01). The rate constants (kT) for patulin degradation with cysteine (0.0036~0.3200 μg/L•min) were far more than those of treatments without cysteine (0.0012~0.1614 μg/L•min), and the activation energy (Ea=43.89 kJ/mol) was far less than that of treatment without cysteine (61.74 kJ/mol). Increasing temperature could obviously improve the degradation efficiency of patulin, regardless of the presence of cysteine. Thus, both cysteine and high temperature decreased the stability of patulin in highly acidic conditions, and improved its degradation efficiency, which could be applied to guide the detoxification of patulin by cysteine in juice processing industry.
(3) Comment: Line 5: What acid was used to adjust the pH? What was the range of cysteine concentrations applied?
Response: In the section of “5.2.1 Preparation of Patulin and Cysteine Solutions”, we have provide the preparation of PCA buffer solution. To simulate the acidic condition of apple juice, pH 3.5 of phosphoric-citric acid (PCA) buffer solutions were firstly prepared by mixing 0.2 mol/L Na2HPO4 solution and 0.1 mol/L citric acid solution with suitable volumes, which were checked using a pH meter (pHs-3C, Leici, China) and adjusted them with 1 mol/L of phosphoric acid or sodium hydroxide solution.
The concentration of cysteine working solution used in this study is 30 μmol/L. The reaction conditions of patulin and cysteine are as followed: 30 μmol/L of cysteine solutions and 10 μmol/L of patulin solutions with a molar ratio of 3:1 (pH 3.5) were added into a 10 mL of reaction kettle, which were quickly mixed and placed in an oil bath at different temperatures (90, 120, and 150℃) to react for different times (0, 30, 60, 90, 120, 150, and 180 min), respectively. It was provided in the section 5.2.2.
(4) Comment: Line 9: What was complexed?
Response: The meaning of “complexed” is that the degradation kinetics of patulin in the absence of cysteine can not be simulated with one model. According to the results of this study, the zero-, first-, and second order models can all fit the degradation kinetics of patulin in the absence of cysteine at different temperatures based on the correlation coefficients (R2>0.90). So we used the word “complexed” to express this meaning.
(5) Comment: Line 13: The authors should include degradation rates. Data should be compared in terms of statistical differences.
Response: In this study, we calculated the rate constants (kT) based on the different degradation kinetic models at different temperatures (See Table 1, Table 2, and Table 3 in our manuscript), which represented the degradation rates of patulin by cysteine. The higher kT values and the greater the degradation rates of patulin by cysteine.
In addition, we compared the data in terms of statistical differences.
“At the same reaction time, cysteine and temperature significantly affected the degradation efficiency of patulin in highly acidic conditions (p<0.01).”
(6) Comment: Line 27: Long sentence. What do the authors consider low molecular weight?
Response: It is indeed a long sentence. To facilitate the reader to understand, we modified the long sentence to short sentences. i.e. Patulin (PAT) is a low-molecular-weight mycotoxin, which has many hazards to the health of human and animals based on the animal and cytotoxicity experiments. These hazards include agitation, vomiting, intestinal hemorrhages, convulsions, ulceration, nausea, and lesions in the duodenum, as well as chronic neurotoxic like genotoxic, teratogenic and immunotoxic effects [1-4].
The molecular weight of patulin (C7H6O4) is 154.12. Compared to the molecular weights of other common mycotoxins, such as aflatoxins (B1:312.27, B2:314.28, G1:328.27, G2:330.29, M1:328.27, M2:330.29), Ochratoxin A (423.67), Deoxynivalenol (296.32), FB1(721.83), T-2 toxin (466.52), its molecular weight is very small. So, many experts call patulin as a low-molecular-weight mycotoxin.
(7) Comment: Line 43: What do the authors consider an ideal method? There have been methods described and these should be cited within the Introduction.
Response: Generally, ideal detoxification methods must: (1) inactive, destroy, or remove the toxin, (2) not produce or leave new toxic substances, (3) retain nutritive value/acceptability of product, (4) not significantly alter the processing technology of product, (5) if possible, destroy fungal spores, and (6) be practical in so far as it is technologically and economically feasible. These requires for an ideal method are listed in our review paper (Diao et al., 2018. Removing and detoxifying methods of patulin: A review. Trends in Food Science & Technology, 81:139-145). In this article, we reviewed the latest development in the removal and detoxification of patulin using physical, chemical, and biological methods, points out their disadvantages, summarizes the degradation products and their safety of patulin, and draws the degradation pathway of patulin.
To avoid duplication in the contents, here we only cite the conclusion in our published papers (Diao et al., 2018), and we think that it is no necessary to list the advantages and disadvantages of the various methods in detail in this study.
(8) Comment: Line 50: Which food is cysteine used? In addition, the authors should comment on the stability of cysteine at alkali pH.
Response: According to our investigation, cysteine is often used in food processing and storage as anti-aging, anti-browning, radical scavenger and reducing agents, such as anti-browning agent in preventing enzymatic browning of “Stanley” plum fruit during cold storage (see reference 16); anti-aging agent for bread (see reference 19); radical scavenger and reducing agents for delaying senescence of harvested longan fruit (Li et al., 2018, L-cysteine hydrochloride delays senescence of harvested longan fruit in relation to modification of redox status. Postharvest Biology and Technology, 143:35-42).
About the stability of cysteine, it must to be considered. So you give us a good idea. It is well known that cysteine as a neutral amino acid, is stable under neutral and acidic conditions, while it is easily oxidized to cystine under alkaline conditions. According to the reported literatures, the Michael addition reaction between patulin and cysteine were performed under weak-acidic or near-neutral conditions. So we modified the sentence (Line 62) to “So it is maybe an idea method in detoxification of patulin in foods with weak-acidic or near-neutral pH values using cysteine.”
(9) Comment: Line 55: Were the preliminary studies published? Is the Patulin-cysteine adducts cytotoxic?
Response: Preliminary studies refer to the pre-experiments, which is not published. In this study, we found that heating with the temperature greater than 80℃ could promote patulin degradation by cysteine under highly acidic conditions based on the results from pre-experiments.
According to the reported literatures (References 14-15), the toxicities of patulin-cysteine adducts are significantly lower than that of patulin itself. However, the degradation products of patulin by cysteine under high acidic and high temperature conditions are still unclear, and their toxicities need to be evaluated in the next study.
(10) Comment: Line 88: The solution only represents apple juice with respect to pH. Therefore, phosphoric acid solutions would be more appropriate.
Response: Our pre-experiments found that patulin is very stable to heat in strong acidic conditions, which is very difficult to remove or degrade by cysteine based on the Michael addition reaction (nucleophilic addition reaction) at room temperature. It is necessary to effectively remove patulin by cysteine with the help of high temperature greater than 90℃ and long reaction time more than 0.5 h (Fig.2 and Fig.3). Therefore, we think that the pH value of foods is one of the critical factors influencing the removal of patulin in foods. Apple juice as an apple based food, which is often contaminated by patulin. Its pH value is in the range from pH 3.0 to 4.5, and usually is around 3.5. In this study, we used pH 3.5 of phosphoric-citric acid (PCA) buffer solution to simulate the acidic condition of apple juice, and the main objectives are to investigate the thermal stability and degradation kinetics of patulin in highly acidic conditions with and without the cysteine. In order to clearly understand the effects of cysteine on the thermal stability of patulin in highly acidic conditions, and to reduce the interference of juice compositions, the study used PCA buffer solution to simulate the acidic environment of apple juice.
(11) Comment: Figure 2: Actual concentration of cysteine would be more informative than % decline. The authors should consider highlighting statistical differences and error bars within the graph. The legend should have sufficient information so that the figure can be understood in isolation.
Response: Firstly, thanks you very much for this suggestion. Figure 2 shows the degradation efficiencies of patulin in pH 3.5 of PCA solution without and with cysteine at different temperatures (90~150℃). In this study, all thermal degradation assays were performed in triplicate, and residual patulin concentrations (μg/L) were shown with means ± standard deviation (SD) (See the experimental data). However, the initial patulin concentration prepared with pH 3.5 of PCA solution had some errors in the experimental processes due to the sampling error and determination error. Therefore, it is difficult to compare the stability of patulin using the residual patulin concentration (μg/L) due to its initial concentration errors. We considered that the degradation efficiencies of patulin (%) can more accurately and scientifically compare the thermal stability of patulin.
The degradation efficiency of patulin was calculated as the followed the formula:
Degradation efficiency (%) = (Mean of patulin before degradation – Mean of residual patulin) ×100/ Mean of patulin before degradation
So, the standard deviation cannot be calculated and without error bars labelled on Figure 2.
(12) Comment: Line 105: What concentration of cysteine?
Response: The concentration of cysteine is 30 μmol/L.
(13) Comment: Line 171-183: This part belongs within the Introduction.
Response: About 3.1 Thermal stability of patulin without and with cysteine (Line 185-203), we place them in the discussion section to facilitate comparison of the thermal stability of patulin reported by other researchers with our results, which provides an opportunity to clearly show the thermal stability of patulin under different experimental conditions to readers. So we think that placing them in the Discussion is better than in the Introduction.
(14) Comment: Line 187: Is it more that cysteine adduct formation was enhanced with temperature rather than decreasing the thermal stability of the mycotoxin. It would have been interesting to compare the reaction of ascorbic acid and cysteine.
Response: According to the results from this study, we are unable to determine what the degradation products of patulin by cysteine are. While the fact is that patulin was significantly reduced by cysteine under the high acidic and high temperature conditions based on the determination of HPLC. So we think that the structure of patulin should be destroyed by cysteine and resulted in its decrease in thermal stability.
As you said, it would have been interesting to compare the reaction of ascorbic acid and cysteine. While we used pH 3.5 of PCA buffer solution to simulate the acidic environment of apple juice in this study, which contains no ascorbic acid, so it can not compare the reaction of ascorbic acid and cysteine. In the next research, we will explore it according to your suggestion.
(15) Comment: Line 199: The authors could only speculate on the mechanisms and should have at least confirmed the structure of the adduct.
Response: Thanks you for this suggestion. About the structures of adducts, we think that they are not the focus of this study. In this study, the main objectives are to investigate the thermal stability and degradation kinetics of patulin in highly acidic conditions with and without the cysteine. The removal mechanisms of patulin by cysteine were only speculated by us based on the studies reported by Fliege & Metzler (2000) and Collin et al. (2008). In the discussion, it is only to explain the reasons about the removal of patulin by cysteine at highly acidic and high temperature conditions.
(16) Comment: Line 267: Is phosphoric acid representative of apple juice? Malic acid would have been more appropriate or even apple juice if this was the system being represented.
Response: According to our investigation and pre-experiments, pH value of foods is one of the critical factors influencing the removal of patulin in foods. In this study, we used pH 3.5 of phosphoric-citric acid (PCA) buffer solution to simulate the acidic condition of apple juice, and the main objectives are to investigate the thermal stability and degradation kinetics of patulin in highly acidic conditions with and without the cysteine.
As you said, the PCA buffer solution can not represent apple juice. In order to clearly understand the effects of cysteine on the thermal stability of patulin in highly acidic conditions, and to reduce the interference of juice compositions, the study only used PCA buffer solution to simulate the acidic environment of apple juice.
(17) Comment: Line 279: Would these temperatures represent those in apple juice processing?
Response: In this study, we explored the thermal stability of patulin in highly acidic conditions at 90, 120, and 150℃, respectively. 90 and 120℃represent pasteurization and high-temperature sterilization in juice processing, respectively. 150℃ is used only to determine the maximum tolerance temperature of patulin.
(18) Comment: Line 292: Was adduct formation determined by the disappearance of patulin or formation of Patulin-cysteine?
Response: Since we have not yet determined what the degradation products are in this study, including the formation of patulin-cysteine adducts, we can only determine the adduct formation by disappearance of patulin using HPLC.
We tried our best to improve the manuscript and made some changes in the manuscript. These changes will not influence the content and framework of the paper. And here we list the changes. We appreciate for your warm work earnestly, and hope that the correction will meet with approval.
Once again, thank you very much for your comments and suggestions.
Reviewer 2 Report
Thank you for the opportunity to review this article.
The manuscript is extremely well written.
Here are some corrections.
Line 31.You can add the recent puplication Agriopoulou, S.; Stamatelopoulou, E.; Varzakas, T. Advances in Occurrence, Importance, and Mycotoxin Control Strategies: Prevention and Detoxification in Foods. Foods 2020, 9, 137.
Line 43.no idea method has been found...Maybe ideal?
Line 204, 211.NH2..Please write NH2
Author Response
Dear Reviewer:
Thank you for the comments concerning our manuscript entitled “Thermal Stability and Degradation Kinetics of Patulin in Highly Acidic Conditions: Impact of Cysteine” (ID: toxins-1364184). Those comments are all valuable and very helpful for revising and improving our paper. We have studied the comments carefully and made corrections which we hope to meet with approval. The main corrections in the paper and the responds to the reviewers’ comments are as follows:
(1) Comment: The manuscript is extremely well written.
Response: Thank you very much for your recognition to our work.
(2) Comment: Line 31. You can add the recent publication Agriopoulou, S.; Stamatelopoulou, E.; Varzakas, T. Advances in Occurrence, Importance, and Mycotoxin Control Strategies: Prevention and Detoxification in Foods. Foods 2020, 9, 137.
Response: According to your suggestion, we add the reference.
(3) Comment: Line 43.no idea method has been found...Maybe ideal?
Response: Generally, ideal detoxification methods must: (1) inactive, destroy, or remove the toxin, (2) not produce or leave new toxic substances, (3) retain nutritive value/acceptability of product, (4) not significantly alter the processing technology of product, (5) if possible, destroy fungal spores, and (6) be practical in so far as it is technologically and economically feasible. These requires for an ideal method are listed in our review paper (Diao et al., 2018. Removing and detoxifying methods of patulin: A review. Trends in Food Science & Technology, 81:139-145).
Theoretically, removal of patulin in foods using cysteine can satisfy the six requirements above-mentioned. However, the degradation products of patulin by cysteine under high acidic and high temperature conditions are still unclear presently, and their toxicities need to be evaluated in the next study.
(4) Comment: Line 204, 211.NH2..Please write NH2
Response: Thanks you for pointing out the writing error in our manuscript, and we revised it.
We tried our best to improve the manuscript and made some changes in the manuscript. These changes will not influence the content and framework of the paper. And here we list the changes. We appreciate for your warm work earnestly, and hope that the correction will meet with approval.
Once again, thank you very much for your comments and suggestions.
Round 2
Reviewer 1 Report
The authors have addressed some issues but not all those raised in the previous review. For example, reference is still made to apple juice even though the studies were only performed in phosphoric acid. Moreover, % reduction was retained within the text when actual amounts of patulin degraded would have been more informative. The Discussion still contains speculative statements with no data generated from the study to support arguments.
Author Response
Dear Reviewer:
Thank you very much for giving us the opportunity to revise our manuscript (ID: toxins-1364184) once again. Your comments are all valuable for improving the quality of paper. We have studied the comments carefully and made corrections which we hope to meet with approval. The main corrections in the paper and the responds to your comments are as follows:
(1) Comment: Reference is still made to apple juice even though the studies were only performed in phosphoric acid.
Response: It is well known that patulin is often contaminated apple juices. The detoxification of patulin in model solutions or apple juices had been widely studied by researchers. In the first response to your comments, we inferred that the pH value of foods is one of the critical factors influencing the removal of patulin based on the reported literatures. Apple juice as an apple based food, its pH value is in the range from pH 3.0 to 4.5, and usually is around 3.5. Therefore, we used pH 3.5 of phosphoric-citric acid (PCA) buffer solution to simulate the acidic condition of apple juice, and the main objectives of this study are to investigate the thermal stability and degradation kinetics of patulin in highly acidic conditions with and without the cysteine. In order to clearly understand the effects of cysteine on the thermal stability of patulin in highly acidic conditions, and to reduce the interference of juice compositions, we used PCA buffer solution to simulate the acidic environment of apple juice.However, in this study, in order to clearly explain the research background and the current situation in detoxifying patulin, we still cite some literatures about patulin detoxification in model solutions and apple juices. These cited literatures do not affect our study purposes and results.
(2) Comment: % reduction was retained within the text when actual amounts of patulin degraded would have been more informative.
Response: Thanks you for the good suggestion. According to your suggestion, we revised the Figure 2, and added the Figure 2A to show the residual patulin concentrations in pH 3.5 of solutions without and with cysteine at different temperatures (90~150℃).
(3) Comment: The Discussion still contains speculative statements with no data generated from the study to support arguments.
Response: We have already considered the questions that you have raised in the Discussion. As you said, Discussion still contains speculative statements with no data generated from the study to support arguments. The speculative statements are the degradation mechanisms of patulin by cysteine under high acidic and high temperature conditions (Line 201-214).
Considering the integrity and readability of Discussion, in this study, we boldly proposed three possible degradation mechanisms of patulin by cysteine under high acid and high temperature, which gives the reader a comprehensive understanding of the degradation process of patulin by cysteine based on our results. However, these speculations all have a scientific theoretical basis according to the results from previous studies.
For example, the first speculative mechanism (The first mechanism remains a nucleophilic addition reaction), is inferred from the Brownian motion or the thermal motion of the molecules, as well as the Reference 14 (Fliege, R.; Metzler, M. Electrophilic properties of patulin. N-acetylcysteine and glutathione adducts. Chem. Res. Toxicol. 2000,13, 373–381.).
The second one is the combination of thermal degradation and nucleophilic addition reaction, which is inferred based on the researches by Fliege & Metzler (2000) (Reference 14) and Collin et al. (2008) (Reference 25: Collin, S.; Bodart, E.; Badot, C.; Bouseta, A.; Nizet, S. Identification of the main degradation products of patulin generated through heat detoxification treatments. J. Inst. Brew. 2008, 114, 167–171.). And We used Figure 7 to show the degradation process of patulin by cysteine.
The third is forming hydrogen bonds and covalent bonds between patulin and cysteine (–SH and –NH2 groups) under high acid and high temperature environment, which is summarized based on the results from the References 27-29.
In addition, at the end of this paragraph, we also pointed out the shortcoming of this study and proposed the next work to further verify these inferred mechanisms based on the analysis of Fourier transform infrared spectra (FTIR), mass spectroscopy (MS), nuclear magnetic resonance (NMR), and thin-layer chromatography (TLC).
We tried our best to improve the manuscript and made some changes in the manuscript. These changes will not influence the content and framework of the paper. And here we list the changes. We appreciate for your warm work earnestly, and hope that the correction will meet with approval.
Once again, thank you very much for your comments and suggestions.
